# Tolerance of *Plasmodium falciparum* mefloquine-resistant clinical isolates to mefloquine-piperaquine with implications for triple artemisinin-based combination therapies

Camille Roesch [1,11], Anna Cosson[2], Melissa Mairet Khedim[1], Nimol Khim[1], Nimol Kloeung[1], Sopheakvatey Ke[1], Sreynet Srun[1], Rotha Eam[1], Chanra Khean[1], Chanvong Kul[1], Lucie Adoux[3], Jean Popovici [1,4], Rithea Leang[5], Pascal Ringwald[6], Frédéric Ariey[7,8], Romain Coppée [2,9,10] & Benoit Witkowski [1,4,11] ✉

Triple artemisinin-based combination therapies (TACTs) have been proposed to delay the emergence of multidrug-resistant *Plasmodium falciparum* by combining two partner drugs with an artemisinin derivative. Among these, mefloquine–piperaquine (MQ–PPQ) is a leading candidate, based on the assumption that simultaneous resistance to both partner drugs would be difficult to develop. Here, we assess the efficacy and resistance potential of MQ–PPQ using Cambodian clinical isolates with distinct resistance profiles. We find that MQ resistance confers significant cross-tolerance to the MQ–PPQ combination, whereas PPQ-resistant and -sensitive strains remain susceptible. Under repeated MQ–PPQ pressure for four months, parasites rapidly acquire MQ–PPQ tolerance, driven by *pfmdr1* amplification. Mechanistic investigations reveal that MQ inhibits PPQ accumulation in a dose-dependent manner, providing a functional explanation for the compromised efficacy of the combination. These findings demonstrate that MQ resistance alone can undermine MQ–PPQ TACT efficacy, calling into question the strategic rationale of this combination and underscoring the need for alternative regimens with a lower risk of resistance selection.

Over the past two decades, malaria control efforts have led to substantial reductions in global incidence and mortality[1]. Central to this progress has been the widespread deployment of artemisinin-based combination therapies (ACTs), which remain the cornerstone of first-line treatments for uncomplicated *Plasmodium falciparum* malaria. However, the emergence of artemisinin (ART) resistance (ART-R) in eastern and central Africa now poses a serious threat to these gains[2]. While partial resistance to ART alone, as previously observed in both Africa and Southeast Asia, does not necessarily result in increased ACT treatment failure

rates, resistance to partner drugs remains the key driver of clinical failure[3,4].

Of particular concern is the potential for ART-R to serve as a precursor to multidrug-resistant parasite lineages. This scenario is supported by previous observations in the Greater Mekong Sub-region, where parasites harboring resistance to at least one partner drug facilitated the subsequent emergence of ACT-resistant strains[5–10]. The increasing detection of ART-R in Africa raises the possibility of a similar trajectory, potentially leading to a widespread loss of ACT efficacy. To mitigate this risk, one alternative strategy is to develop optimized ACT formulations with a lower propensity for resistance selection, even in the presence of pre-existing drug resistance.

Triple ACTs (TACTs) have been proposed as a potential solution to this challenge. By combining an ART derivative with two partner drugs, TACTs are designed to reduce the likelihood of resistance selection and maintain therapeutic efficacy in regions where conventional ACTs are failing. Two main formulations have been evaluated in clinical trials: artesunate–mefloquine–piperaquine and artesunate–amodiaquine–lumefantrine[11–14]. Among these, mefloquine–piperaquine (MQ–PPQ) has drawn particular interest due to a possible antagonism in the acquisition of resistance to its two partner drugs. Data from Southeast Asia suggest that resistance markers for MQ (*pfmdr1* amplification) and PPQ (*pfplasmepsin2/3* (*pfpm2/3*) amplification) display inverse prevalence trends, depending on whether PPQ- or MQ-based ACTs are implemented[15]. Furthermore, MQ–PPQ TACT has demonstrated high efficacy in patients infected with DHA–PPQ-resistant parasites, with cure rates exceeding 95%, compared to 48% for DHA–PPQ alone[14]. This finding highlights MQ's ability to compensate for PPQ resistance in the context of pre-existing PPQ resistance.

However, the potential of MQ–PPQ TACT in the context of pre-existing MQ resistance remains unexplored. Given its possible large-scale implementation, addressing this gap is crucial to ensure its long-term viability. In this study, we aimed to evaluate the efficacy of MQ–PPQ TACT against *P. falciparum* isolates exhibiting MQ resistance, providing critical insights into its operational feasibility and potential limitations.

## Results

### In vitro susceptibility of field isolates to PPQ, MQ and combined MQ–PPQ exposure

The in vitro susceptibility of a panel of Cambodian *P. falciparum* isolates, categorized as sensitive (Sensitive, n = 5), PPQ-resistant (KEL1/PLA1, n = 9) and MQ-resistant (MQ-R, n = 17), was first evaluated for PPQ, MQ, and the MQ–PPQ combination. Based on the Piperaquine Survival Assay (PSA), KEL1/PLA1 isolates exhibited higher survival rates (PSA 55.58% ± 15.67 SD) compared to both Sensitive (PSA 0.00% ± 0.00 SD, $p = 0.0003$, Kruskal–Wallis test followed by Dunn's multiple comparison test) and MQ-R parasites (PSA 0.16% ± 0.37 SD, $p = 5.00 \times 10^{-5}$) (Fig. 1A).

As expected, the mean half-maximal inhibitory concentration (IC$_{50}$) values determined by [³H]-hypoxanthine incorporation differed significantly between the three groups. MQ-R isolates exhibited significantly higher IC$_{50}$ values for MQ (96.35 nM ± 22.84 SD) than both Sensitive and KEL1/PLA1 parasites (42.99 nM ± 14.60 SD, $p = 3.70 \times 10^{-5}$ and 30.31 nM ± 15.13 SD, $p = 3.00 \times 10^{-8}$, respectively, ordinary one-way ANOVA followed by Tukey's multiple comparison test) (Fig. 1B).

Finally, the susceptibility of the three groups to MQ–PPQ co-exposure was assessed using a survival assay across a range of drug concentrations, allowing for Survival half-maximal inhibitory concentration (Survival IC$_{50}$) determination. This methodology was chosen for MQ–PPQ in preference to the conventional [³H]-hypoxanthine uptake inhibition assay, given published data showing the difficulty to generate reliable dose effect curves with PPQ against KEL1/PLA1

strains[4]. MQ-R isolates showed significantly higher Survival IC$_{50}$ values for MQ-PPQ (51.26 nM ± 16.29 SD) compared to both Sensitive and KEL1/PLA1 isolates (23.20 nM ± 2.75 SD, $p = 5.00 \times 10^{-6}$ and 24.29 nM ± 8.02 SD, $p = 2.40 \times 10^{-5}$, respectively, ordinary one-way ANOVA followed by Dunnett's T3 multiple comparison test) (Fig. 1C). To evaluate whether MQ resistance is linked to MQ–PPQ susceptibility, the correlation between IC$_{50}$ values of MQ (determined by [³H]-hypoxanthine incorporation) and the Survival IC$_{50s}$ under MQ–PPQ co-exposure was assessed. The result showed a high and significant correlation ($r = 0.7347$, $p = 2.53 \times 10^{-6}$, Spearman correlation test) (Fig. 1D).

### Paired MQ–PPQ in vitro drug pressure

To investigate the potential for stepwise adaptation to MQ–PPQ combination therapy, the Cambodian *P. falciparum* 9097 strain was subjected to four successive rounds of in vitro drug pressure, as summarized in Supplementary Fig. 1. Then, the in vitro susceptibilities of the parental strain and the final selected strain (*Pressure 4*) were compared for each drug individually and for the MQ–PPQ combination. To enable a more accurate comparison of MQ, PPQ, and MQ–PPQ susceptibility shifts, the same methodology based on survival assay yielding Survival IC$_{50}$ was chosen. Following drug pressure, the selected strain exhibited a significantly increased Survival IC$_{50}$ for MQ (146.60 nM ± 33.76 SD vs. 63.69 nM ± 19.37 SD, $p = 0.0053$, unpaired $t$-test) (Fig. 2A), suggesting an adaptive response to MQ exposure. In contrast, no significant change in PPQ susceptibility was observed (Survival IC$_{50}$ 29.50 nM ± 10.19 SD vs. 40.59 nM ± 16.29 SD, $p = 0.2546$) (Fig. 2B), indicating that PPQ susceptibility remained largely unaffected by MQ–PPQ selective pressure.

Interestingly, the Survival IC$_{50}$ for MQ–PPQ co-exposure increased significantly from 21.70 nM ± 4.41 SD to 45.11 nM ± 15.82 SD ($p = 0.0291$) (Fig. 2C), suggesting that adaptation to MQ–PPQ treatment primarily involved increased MQ tolerance rather than PPQ resistance.

The in vitro susceptibility of both parental and selected strains to MQ and PPQ was further assessed using conventional methods, including the [³H]-hypoxanthine uptake inhibition assay and PSA. The MQ IC$_{50}$ increased significantly from 48.37 nM ± 8.68 SD to 117.00 nM ± 18.87 SD ($p = 4.72 \times 10^{-6}$, unpaired $t$-test), confirming the emergence of an MQ resistance phenotype. A modest but significant increase in PSA survival was also observed (1.42% ± 1.34 SD vs 3.35% ± 0.79 SD, $p = 0.0068$, Mann-Whitney $U$ test), although the survival rate remained below the established 10% resistance threshold (Supplementary Fig. 2).

To explore potential genetic mechanisms underlying these phenotypic changes, we assessed copy number variation (CNV) in key resistance-associated genes. The *Pressure 4* strain acquired an additional copy of the *pfmdr1* gene compared to the parental strain (1.96 ± 0.14 SD, n = 9 technical replicates), a known marker of MQ resistance. However, no change was detected in *pfpm2* copy number (0.93 ± 0.08 SD, n = 9 technical replicates), suggesting that adaptation to MQ–PPQ was primarily mediated by *pfmdr1* amplification (Supplementary Table 1).

### Comparative genomic analysis of parental and selected strains

To validate our previous findings, we performed whole-genome sequencing of both the parental and *Pressure 4 P. falciparum* strains. Comparative genomic analysis revealed no newly acquired single nucleotide polymorphisms (SNPs) or indels following drug selection, suggesting the absence of specific point mutations that could directly explain the increased tolerance to the MQ–PPQ combination (Supplementary Fig. 3). Notably, no mutations were detected in *pfcrt*, a well-characterized marker of PPQ resistance.

In contrast, CNV analysis identified an amplification of a region on chromosome 5 (801,000–970,000), encompassing *pfmdr1*, in the *Pressure 4* strain (Fig. 3 and Supplementary Table 2). No concomitant amplification of *pfpm2* or *pfpm3* was observed. Additionally, a

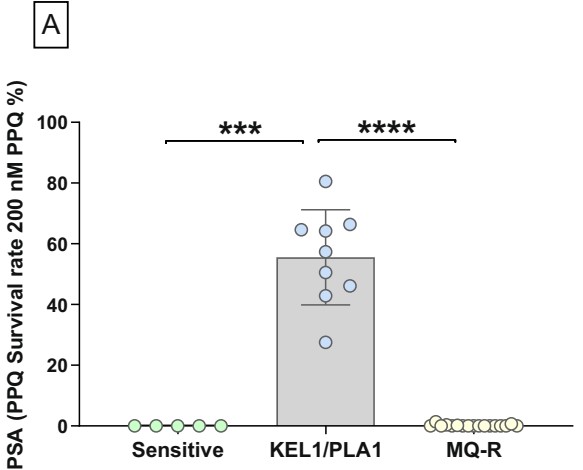

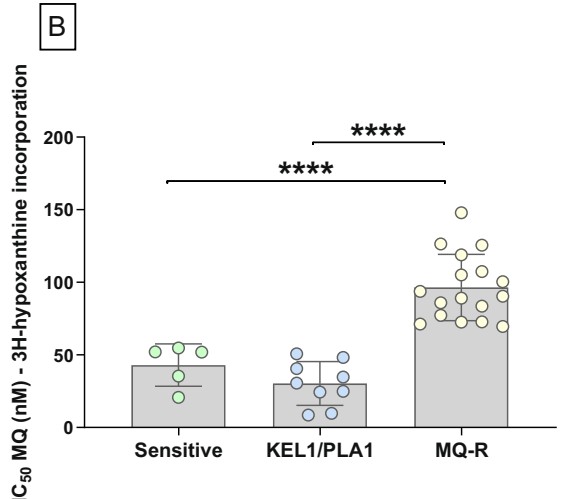

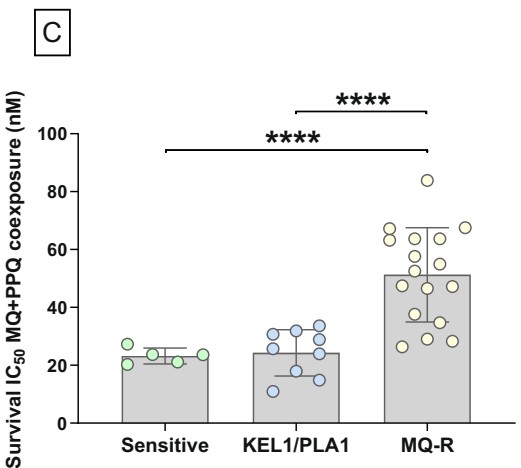

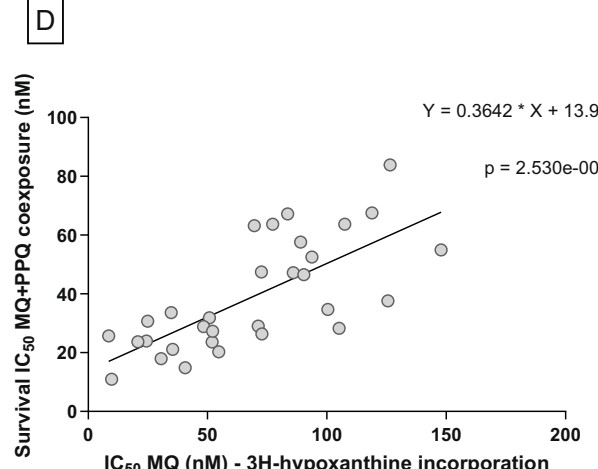

**Fig. 1 | Susceptibility assessment of isolates groups to PPQ, MQ and MQ-PPQ.** In vitro susceptibility of sensitive (sensitive, n = 5, green), PPQ-resistant (KEL1/PLA1, n = 9, blue) and MQ-resistant (MQ-R, n = 17, yellow) parasites to PPQ (**A**), MQ (**B**), and paired MQ–PPQ (**C**). Mean and standard deviation are shown on the graphs. Kruskal–Wallis followed by Dunn's multiple comparison test or ordinary one-way ANOVA followed by Tukey's or Dunnett's T3 multiple comparison tests were used for statistical analysis. P-values are based on two-sided tests. **A** Susceptibility of clinical isolates to PPQ measured by PSA. KEL1/PLA1 isolates have a significantly higher survival rate than both Sensitive (*** $p = 0.0003$) and MQ-R parasites (**** $p = 5.00 \times 10^{-5}$). Shapiro–Wilk test confirmed normality for the KEL1/PLA1 group ($p = 0.9503$) but not for MQ-R group (**** $p = 2.00 \times 10^{-6}$). **B** Susceptibility of clinical isolates to MQ measured by [³H]-hypoxanthine incorporation. MQ-R isolates have significantly higher $IC_{50}s$ than both Sensitive (**** $p = 3.70 \times 10^{-5}$) and KEL1/PLA1 parasites (**** $p = 3.00 \times 10^{-8}$). Shapiro–Wilk test p-values were 0.1370 (Sensitive), 0.6103 (KEL1/PLA1), and 0.1747 (MQ-R). **C** Susceptibility of clinical isolates to PPQ and MQ co-exposure. Survival $IC_{50}s$ were calculated after measuring the survival rate of parasites at increasing concentrations of MQ–PPQ combination. MQ-R isolates exhibited significantly higher Survival $IC_{50}s$ than both Sensitive (**** $p = 5.00 \times 10^{-6}$) and KEL1/PLA1 parasites (**** $p = 2.40 \times 10^{-5}$). Shapiro–Wilk test p-values were 0.5980 (Sensitive), 0.4375 (KEL1/PLA1), and 0.5649 (MQ-R). **D** Correlation between MQ $IC_{50}$ and MQ–PPQ Survival $IC_{50}$. The $IC_{50}s$ of MQ and the Survival $IC_{50}s$ of the combination MQ–PPQ exposure were highly correlated ($r = 0.7347$, **** $p = 2.53 \times 10^{-6}$ Spearman correlation test, Shapiro-Wilk test for normality gave * $p$ value = 0.0350). Source data are provided as a Source Data file.

deleted genomic region was identified on chromosome 9 (1,377,500–1,473,500) following drug selection. This deletion is unlikely to be involved in drug resistance acquisition, as similar deletions have been reported in long-term in vitro cultures of *P. falciparum*[16]. Altogether, the comparative genomic analysis suggests that *pfmdr1* amplification may play a pivotal role in modulating parasite susceptibility to MQ–PPQ, potentially by altering drug import or efflux mechanisms within the parasite.

### Evaluation of the central role of *pfmdr1* in MQ–PPQ tolerance using radioactive assays

To investigate the potential involvement of *pfmdr1* in tolerance to the MQ–PPQ combination, we assessed the impact of MQ pre-incubation

on the intracellular incorporation of radiolabeled PPQ-[³H(G)]. In the *Pressure 4* strain, the amount of incorporated radiolabeled PPQ was significantly reduced when parasites were pre-exposed to MQ before the addition of radiolabeled PPQ (16.98 ± 2.63 SD vs. 4.64 ± 2.03 SD, $p = 3.00 \times 10^{-6}$, paired *t*-test) (Fig. 4A). A similar effect was observed in the parental strain, where MQ pre-incubation resulted in a significant decrease in PPQ incorporation (14.54 ± 3.99 SD vs. 1.82 ± 1.83 SD, $p = 1.60 \times 10^{-4}$) (Fig. 4A). These findings suggest that PPQ uptake may be influenced by *pfmdr1*-mediated drug transport.

To further explore this hypothesis, we measured variation in PPQ uptake among clinical isolates with different resistance phenotypes (Sensitive, n = 5, MQ-R, n = 13 and KEL1/PLA1, n = 7). In all three groups, MQ pre-incubation significantly reduced PPQ uptake (Fig. 4B). The

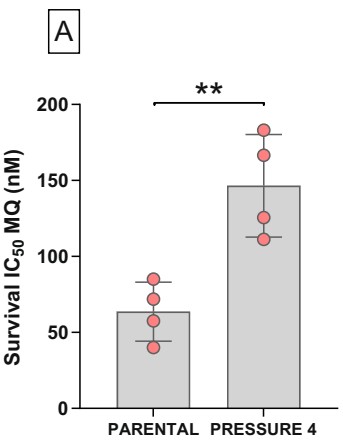

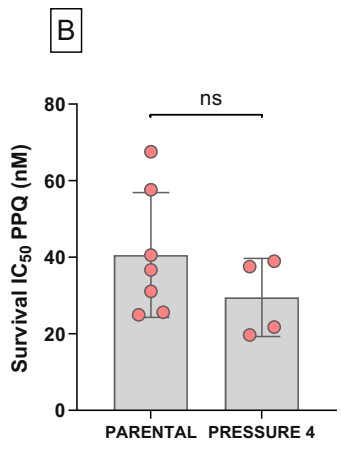

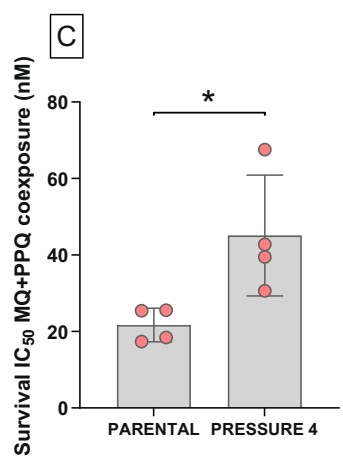

**Fig. 2 | Evolution of in vitro susceptibility in the parental strain and the selected strain after continuous exposure to MQ + PPQ.** Each pink dot represents the mean of four independent replicates. Error bars represent standard deviations. P-values are based on two-sided tests. All data were generated using a modified version of PSA, where 0–3 h ring-stage parasites were exposed to increasing concentrations of drug (0–800 nM). Drug was washed after 48 hours, and the experiment was stopped at 72 hours. **A** Evolution of MQ susceptibility (MQ Survival $IC_{50s}$). The susceptibility of the selected strain to MQ decreased (**$p = 0.0053$, Unpaired $t$-test–Shapiro-Wilk test, $p$-value = 0.9602 and 0.5519, respectively for *Parental* and *Pressure 4*). **B** Evolution of PPQ susceptibility (PPQ Survival $IC_{50s}$). The susceptibility to PPQ of the selected strain did not change ($p = 0.2645$, unpaired $t$-test, $p$-value = 0.2607 and 0.1211, respectively for *Parental* and *Pressure 4*). **C** Evolution of MQ–PPQ co-exposure susceptibility (MQ + PPQ Survival $IC_{50s}$). The susceptibility to MQ–PPQ co-exposure of the selected strain decreased (*$p = 0.0291$, unpaired $t$-test – Shapiro-Wilk test, $p$-value = 0.0910 and 0.4077, respectively for *Parental* and *Pressure 4*). Source data are provided as a Source Data file. ns: non significant.

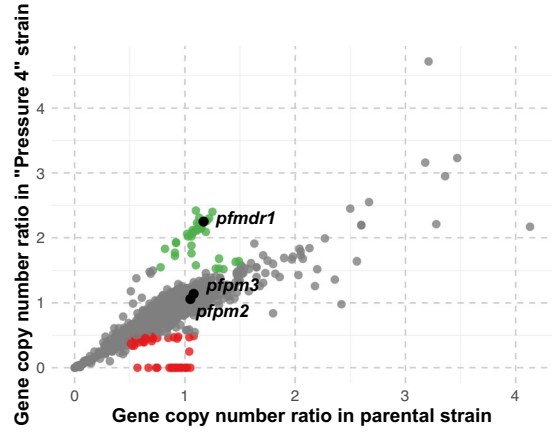

- ● Gene amplified in "Pressure 4" strain   ● Gene deleted in "Pressure 4" strain

**Fig. 3 | Gene copy number variations between the parental and pressure 4 strains.** Each dot represents a gene. Green and red dots indicate genes that are amplified (ratio ≥1.5) or deleted (ratio ≤0.5) in the *Pressure 4* strain, respectively. Grey dots show genes with relatively stable copy numbers. *pfmdr1*, *pfpm2* and *pfpm3*, which have previously been associated with drug resistance, are labeled. The complete list of amplified and deleted genes in the *Pressure 4* strain is provided in Supplementary Table 2. Source data are provided as a Source Data file.

mean radioactive PPQ incorporation decreased from $4.72 \pm 1.16$ SD to $1.12 \pm 1.80$ SD ($p = 0.0625$, Wilcoxon matched-pairs signed rank test) in Sensitive isolates, from $4.65 \pm 1.52$ SD to $0.28 \pm 0.25$ SD ($p = 0.0001$, paired $t$-test) in KEL1/PLA1 isolates, and from $6.84 \pm 2.42$ SD to $1.75 \pm 0.90$ SD ($p = 0.0002$, Wilcoxon matched-pairs signed rank test) in MQ-R isolates.

To quantify the impact of increasing MQ concentrations on PPQ incorporation, we performed a dose-response experiment using pre-incubation with MQ at concentrations ranging from 25 nM to 750 nm prior to the addition of PPQ-[³H(G)] (Fig. 5A). The concentration of MQ required to inhibit 50% of PPQ incorporation was lowest in KEL1/PLA1 (15.57 nM ± 4.13 SD) and Sensitive isolates (33.52 nM ± 5.54 SD), whereas MQ-R parasites required a higher MQ concentration (115.33 nM ± 63.12 SD) to achieve the same level of

inhibition ($p = 0.0312$ and $p = 0.0008$, respectively, Mann-Whitney $U$ test).

Finally, we examined the relationship between *pfmdr1* copy number and PPQ uptake. Isolates with multiple *pfmdr1* copies (n = 10) exhibited significantly higher incorporation of PPQ-[³H(G)] compared to single-copy isolates (n = 12), even when pre-incubated with 200 nM MQ ($p = 0.0160$, Mann-Whitney $U$ test) (Fig. 5B). These findings reinforce the hypothesis that *pfmdr1* plays a central role in modulating the intracellular balance of MQ and PPQ.

**MQ pre-incubation reduces PPQ susceptibility in MQ-R isolates**

Given the observed interaction between MQ and PPQ uptake, we next assessed whether MQ pre-incubation could modulate PPQ susceptibility in MQ-R parasites. Cambodian clinical isolates classified as MQ-R (n = 12), as well as the parental (6 independent replicates) and *Pressure 4* (7 independent replicates) laboratory strains, were subjected to the PSA under standard conditions and following pre-incubation with 200 nM MQ. PSA survival rates varied according to genotype (Fig. 6). The addition of MQ had no significant impact on PSA levels in the parental strain ($p = 0.4688$, Wilcoxon matched-pairs signed rank test). However, in the *Pressure 4* strain, MQ pre-incubation resulted in a significant increase in survival rate, from $3.35 \pm 0.79$ SD to $6.38 \pm 2.06$ SD ($p = 0.0038$, paired $t$-test). A similar effect was observed for MQ-R parasites, where pre-incubation with MQ significantly increased PSA survival from $0.29 \pm 0.37$ SD to $4.36 \pm 4.79$ SD ($p = 0.0024$, Wilcoxon matched-pairs signed rank test). These findings provide further evidence that MQ exposure influences PPQ susceptibility, particularly in MQ-R isolates.

## Discussion

The primary objective of this study was to assess the activity of the MQ–PPQ combination against the different resistance profiles frequently observed in Southeast Asia. In vitro susceptibility testing confirmed that sensitive (characterized by *pfk13* WT, *pfmdr1* monocopy, *pfpm2* monocopy, and no mutations at *pfcrt* codons known to modulate PPQ susceptibility) and KEL1/PLA1 parasites were highly susceptible to MQ, whereas sensitive and MQ-R parasites remained susceptible to PPQ. Conversely, KEL1/PLA1 and MQ-R strains displayed increased tolerance to PPQ and MQ, respectively.

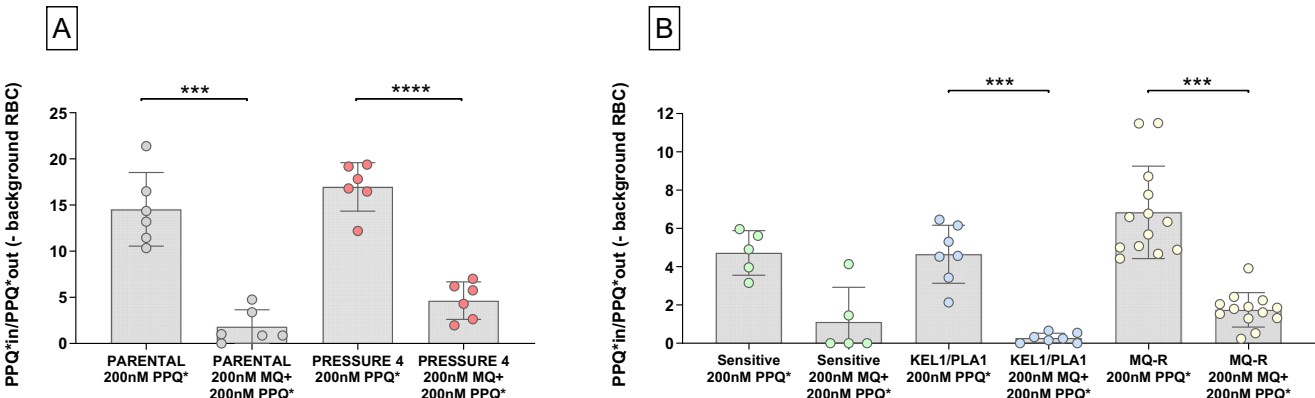

**Fig. 4 | Radioactive PPQ incorporation assay.** Means and standard deviations are represented on the graphs; each dot represents an independent biological replicate. The incorporation of PPQ-[$^3$H(G)] was measured in synchronized trophozoites (21–24 h post-invasion). P-values are based on two-sided tests. **A** Variation in radioactive PPQ incorporation with or without pre-incubation with 200 nM MQ prior to the addition of PPQ-[$^3$H(G)]. Comparison between the two conditions within the parental strain and within the selected strain showed a significant decrease of PPQ incorporation when MQ was pre-incubated (*** $p = 1.60 \times 10^{-4}$ and **** $p = 3.00 \times 10^{-6}$, respectively for *parental* (6 independent replicates, grey) and *pressure 4* (6 independent replicates, pink) strains (paired *t*-test). Shapiro-Wilk test *p*-values were as follows: 0.5932, 0.1454, 0.2227, and 0.5613, respectively for *Parental* 200 nM PPQ*, *Parental* 200 nM MQ + 200 nM PPQ*, *Pressure 4* 200 nM PPQ*,

and *Pressure 4* 200 nM MQ + 200 nM PPQ*. **B** Variation in PPQ incorporation among clinical isolates with different phenotypes. A total of 5, 7, and 13 biological replicates were used for the sensitive (green), KEL1/PLA1 (blue), and MQ-R (yellow) groups, respectively. In all cases, pre-incubation with 200 nM MQ before addition of PPQ-[$^3$H(G)] decreased the incorporation of PPQ by parasites. Statistical significance was achieved for MQ-R and KEL1/PLA1 parasites (*** $p = 0.0002$ and *** $p = 0.0001$, Wilcoxon matched-pairs signed rank test and paired *t*-test, respectively). Shapiro-Wilk test *p*-value = 0.7334, 0.0245, 0.0227, 0.3178, 0.7749 and 0.4900, respectively for Sensitive 200 nM PPQ*, Sensitive 200 nM MQ + 200 nM PPQ*, MQ-R 200 nM PPQ*, MQ-R 200 nM MQ + 200 nM PPQ*, KEL1/PLA1 200 nM PPQ* and KEL1/PLA1 200 nM MQ + 200 nM PPQ*. Source data are provided as a Source Data file. RBC: red blood cell.

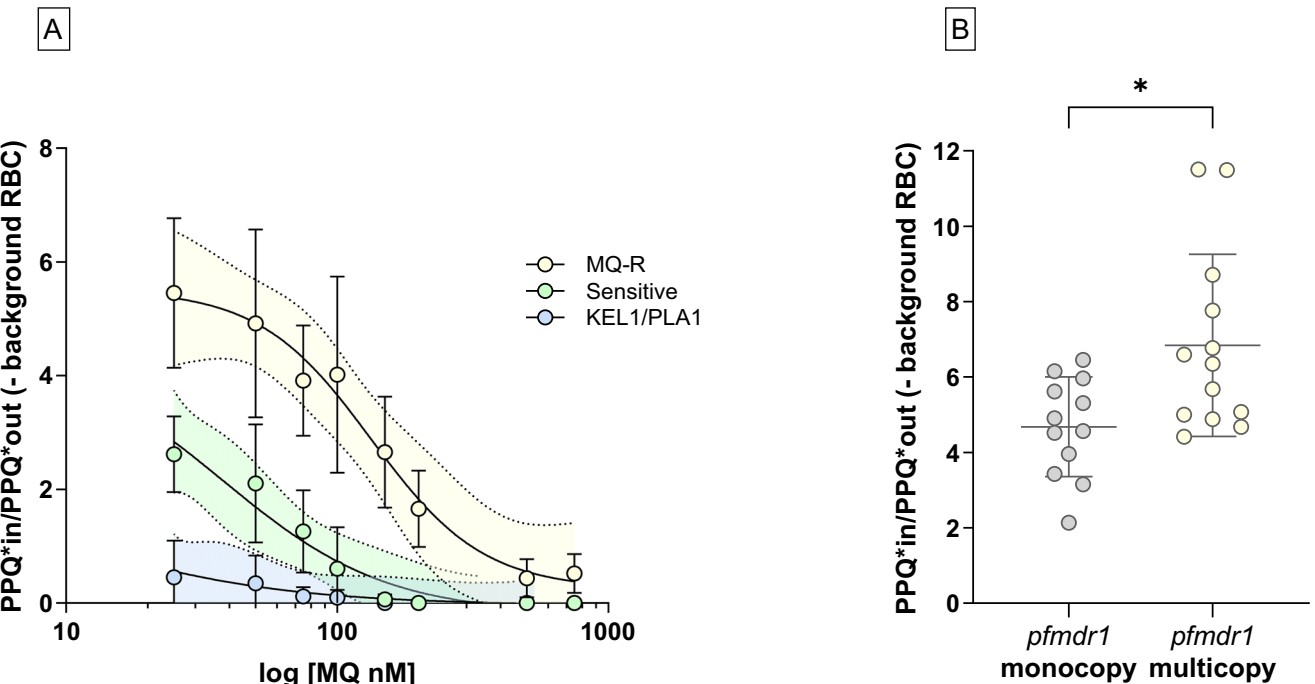

**Fig. 5 | Incorporation of radioactive PPQ among clinical isolates.** The incorporation of PPQ-[$^3$H(G)] was measured in synchronized trophozoites (21–24 h post-invasion). **A** Variation in radioactive PPQ incorporation following pre-incubation of increasing MQ concentrations (25 nM to 750 nM) before addition of PPQ-[$^3$H(G)]. Means and standard deviations are represented on the graphs; each dot represents an independent biological replicate. The solid line represents a non-linear regression between MQ concentration and PPQ uptake; the shaded area indicates the 95% confidence interval (Sensitive, n = 2, green; KEL1/PLA1, n = 2, blue and MQ-R,

n = 5, yellow). **B** Difference in radioactive PPQ incorporation between isolates carrying single vs multiple copies of *pfmdr1*. Isolates with *pfmdr1* amplifications (n = 13 biological replicates, yellow) incorporated significantly more PPQ-[$^3$H(G)] than single-copy isolates (n = 12 biological replicates, grey), even when PPQ-[$^3$H(G)] was added after 200 nM of MQ pre-incubation (* $p = 0.016$, Mann-Whitney $U$ test−Shapiro-Wilk test *p*-value = 0.8337 and 0.0227, respectively for *pfmdr1* monocopy and *pfmdr1* multicopy). Data are presented as mean values +/- SD. P-values are based on two-sided tests. Source data are provided as a Source Data file.

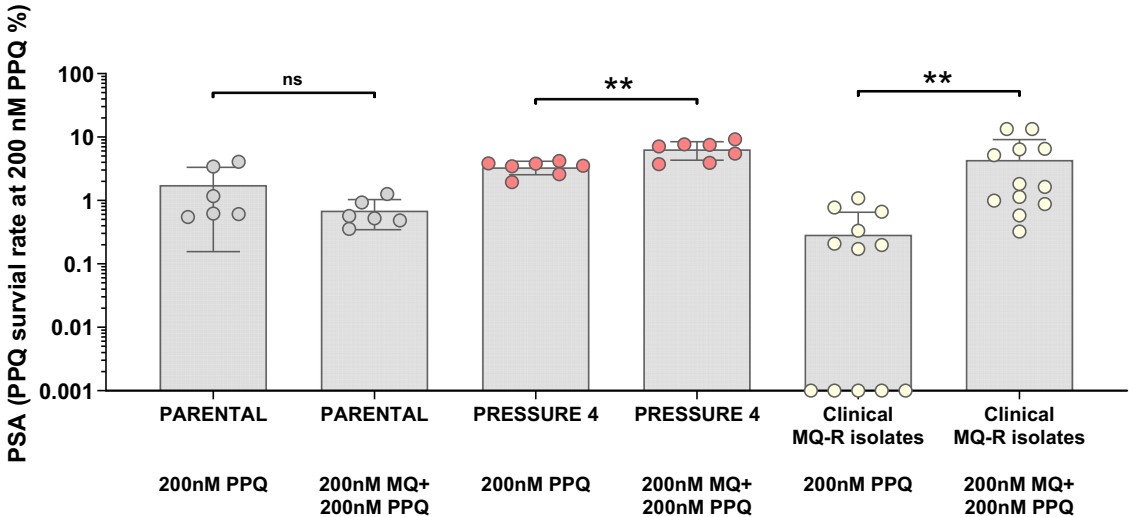

**Fig. 6 | Comparison of the survival rate of the parental line, the selected strain, and clinical isolates with MQ resistance, in response to PPQ with or without MQ pre-incubation.** Means and standard deviations are shown on the graphs; each dot represents an independent replicate. P-values are based on two-sided tests. The median survival of parasites to 200 nM PPQ is unchanged in presence of MQ pre-incubation before exposure to PPQ in the parental strain ($p = 0.4688$, Wilcoxon matched-pairs signed rank test, n = 6 biolocal replicates, grey). The pre-incubation with MQ significantly increased the survival of the selected strain (** $p = 0.0038$, paired $t$-test, n = 7 biological replicates, pink) and of clinical MQ-R parasites (** $p = 0.0024$, Wilcoxon matched-pairs signed rank test, n = 12 biological replicates, yellow). Shapiro-Wilk test $p$-value were 0.0290, 0.2313, 0.2989, 0.4944, 0.0095, and 0.0054, respectively for *Parental* 200 nM PPQ, *Parental* 200 nM MQ + 200 nM PPQ, *Pressure 4* 200 nM PPQ, *Pressure 4* 200 nM MQ + 200 nM PPQ, Clinical MQ-R isolates 200 nM PPQ, and Clinical MQ-R isolates 200 nM MQ + 200 nM PPQ. Zeros are shown as 0.001 for log scale purposes. Source data are provided as a Source Data file. ns: non significant.

Evaluation of the combined MQ–PPQ activity against sensitive and KEL1/PLA1 parasites demonstrated high susceptibility, consistent with clinical studies reporting excellent therapeutic efficacy of this combination in infections with similar parasite genotypes[14,17]. In sharp contrast, MQ–PPQ activity against MQ-R parasites revealed significantly higher tolerance compared to sensitive and KEL1/PLA1 strains. This result contrasts with the proposed rationale for TACTs, which assumes that the presence of two partner drugs should mitigate resistance to either of them[18]. A possible explanation for this unexpected finding could be the presence of parasites with co-existing resistance to both PPQ and MQ, as previously observed in Cambodia[19]. However, our data ruled out this hypothesis, as all MQ-R strains retained full susceptibility to PPQ when tested individually.

A key observation from our study was that MQ–PPQ tolerance in MQ-R parasites was directly correlated with their intrinsic MQ resistance levels. This was confirmed by a strong and significant correlation between MQ IC$_{50s}$ determined by [$^{3}$H]-hypoxanthine incorporation and MQ–PPQ Survival IC$_{50}$ values. Based on these findings, we hypothesized that *pfmdr1* gene amplification – previously associated with MQ resistance – plays a central role in the acquisition of a tolerance phenotype to paired MQ–PPQ. To test this hypothesis, we evaluated the selection potential of MQ–PPQ in a parasite lineage that had previously exhibited intra-host acquisition of MQ resistance. Upon repeated MQ–PPQ exposure for four months, parasites acquired *pfmdr1* amplification, which was associated with increased tolerance to both MQ and MQ–PPQ. In contrast, no known molecular markers or phenotypic evidence of PPQ resistance were observed. Whole-genome analysis of the parental and derived strains confirmed amplification of a region on chromosome 5, encompassing *pfmdr1* among other genes, as the only genomic change. Interestingly, the *Pressure 4* strain harbors an amplification spanning positions 801,000 to 970,000 on chromosome 5, which encompasses the previously described region from 825,600 to 888,300 described by Eastman et al. (Supplementary Fig. 4)[20].

Our findings diverge in several aspects from those reported by Eastman et al. The *Pressure 4* lineage exhibited a moderate but significant reduction in PPQ susceptibility by PSA (with a mean value well below the resistance threshold) and no significant change in Survival IC$_{50}$s, whereas Eastman et al. described a dramatic ~100-fold increase in IC$_{50}$. Similarly, while Eastman et al. observed a pronounced increase in the susceptibility to MQ, our data point to an important decrease in MQ susceptibility for *Pressure 4*. Eastman et al. also reported a substantial decrease in PPQ accumulation in resistant strains, which we did not observe when comparing the *Parental* and *Pressure 4* lines. At the genetic level, the amplification on chromosome 5 of *Pressure 4* also includes *pfmdr1*, while it was de-amplified in Eastman et al. Furthermore, Eastman et al. also observed the acquisition of a mutation in *pfcrt*, while the *pfcrt* sequence in *Pressure 4* remains unchanged. Considering the major role of *pfcrt* and *pfmdr1* in PPQ resistance, these differences may account for some or all of the phenotypic discrepancies observed. Taken together, these discrepancies suggest that, at this stage, the contribution of the 825,600 – 888,300 amplification to the MQ–PPQ tolerance phenotype of *Pressure 4* remains unresolved. Importantly, this genetic change has so far been detected only under in vitro drug pressure and has not been associated with clinical PPQ resistance in Southeast Asia[4]. Moreover, the chromosome 5 sequence analysis from 4 recent PPQ-R strains isolated in Cambodia (2019) did not show such amplification (Supplementary Fig. 4).

Our results strongly support the role of *pfmdr1* amplification in MQ–PPQ tolerance and suggest that PPQ does not effectively reach its intracellular target in these parasites. The acquisition of *pfmdr1* amplification upon MQ–PPQ pressure was relatively rapid, consistent with previous observations of in vitro resistance selection under MQ pressure. Although we did not evaluate the fitness cost or persistence of the amplification detected in this study, it is likely to be similar to what Preechapornkul et al. observed: a moderate fitness disadvantage of *pfmdr1* amplification that might require up to six months to disappear in vitro without drug pressure[21].

The hypothesis of MQ–PPQ tolerance mediated by *pfmdr1* amplification was further supported by radioactive PPQ incorporation assays. Our results showed that PPQ uptake into infected erythrocytes was directly dependent on *pfmdr1* copy number, with

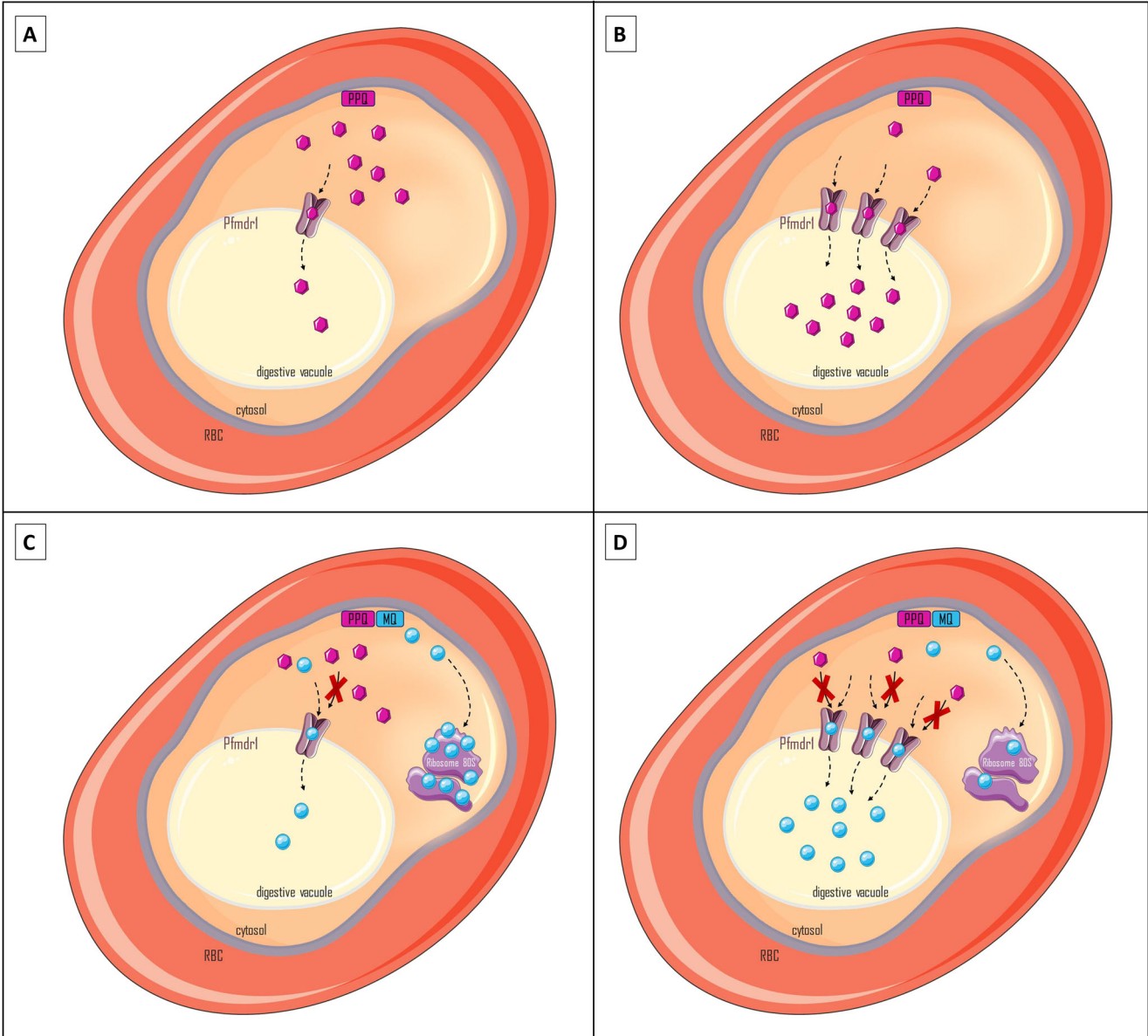

**Fig. 7 | Hypothesis supporting the mechanism of resistance of *P. falciparum* to paired MQ–PPQ by PfMDR1. A** PPQ against *pfmdr1* monocopy lineage. PPQ concentrates in the digestive vacuole through the PfMDR1 channel and likely acts by inhibiting the heme biomineralization process. **B** PPQ against *pfmdr1* multicopy lineage. PPQ accumulates in the digestive vacuole through the PfMDR1 channel at much higher levels than in the previous case, which may explain why parasites with *pfmdr1* amplification exhibit increased sensitivity to PPQ. **C** Paired MQ–PPQ against *pfmdr1* monocopy lineage. A significant amount of MQ reaches the 80S ribosomal subunit and inhibits the parasite development, while a limited quantity is sequestered in the digestive vacuole through PfMDR1 mediated import. The concomitantly present PPQ cannot efficiently reach its site of action due to the competition with MQ at PfMDR1 level. **D** Paired MQ–PPQ against *pfmdr1* multicopy lineage. A significant amount of MQ is sequestered in the digestive vacuole due to PfMDR1 overexpression, resulting in a reduced amount reaching the 80S ribosomal subunit leading to decreased parasite susceptibility. The concomitantly present PPQ cannot efficiently reach its site of action due to the competition with MQ at PfMDR1 level. Image adapted from Servier Medical Art (https://smart.servier.com/), licensed under CC BY 4.0 (https://creativecommons.org/licenses/by/4.0/).

*pfmdr1*-amplified parasites exhibiting significantly higher intracellular PPQ concentrations. This phenomenon explains the observed higher susceptibility of MQ-R parasites to PPQ, highlighting PfMDR1 as a key determinant of PPQ intracellular accumulation and activity. Furthermore, pre-incubation with increasing concentrations of MQ resulted in a dose-dependent reduction in PPQ uptake, suggesting that MQ, a known PfMDR1 substrate, competitively inhibits PPQ uptake. This competition likely reduces intracellular PPQ concentrations to subtherapeutic levels, thereby impairing its efficacy in MQ-R parasites. This experiment and its conclusions are comparable to the findings of Reiling and Rohrbach, who observed a reduction in

Fluo-4 accumulation within the digestive vacuole (DV) in the presence of MQ[22].

The antagonism between MQ and PPQ could be explained by their respective mode of action and mechanism of resistance. The entry of PPQ into the DV may be critical for its activity, whereas MQ entry into the DV is likely to be a detoxification pathway that reduce its cytosolic concentration able to target the 80S ribosomal subunit[23,24]. Our data indicate that this PfMDR1-mediated competition favors MQ, positioning it as the dominant active compound within the MQ–PPQ combination, while PPQ activity is constrained by reduced accumulation.

This hypothesis reconciles the apparent contradiction between the loss of MQ–PPQ activity against *pfmdr1*-amplified parasites and the strong activity of PPQ alone on the same parasites in the absence of MQ. This theory is also consistent with the high clinical efficacy of MQ–PPQ against parasites carrying a single *pfmdr1* copy and PPQ-R isolates from the Mekong region, which may simply reflect the sole antimalarial contribution of MQ. Our hypothesis is summarized in Fig. 7.

In conclusion, this study demonstrates that the core rationale of MQ–PPQ TACT, which relies on antagonistic resistance mechanisms between partner drugs, can inadvertently lead to impaired drug action due to altered intracellular pharmacodynamic. This phenomenon facilitates the emergence of paradoxical resistance driven by *pfmdr1* amplification. Although these findings are based on in vitro observations, they underscore the need for active surveillance of *pfmdr1*-amplified parasites following MQ–PPQ TACT implementation.

## Methods

### Clinical isolate selection and culture conditions

Clinical isolates were obtained from therapeutic efficacy studies (TES) conducted in Cambodia between 2016 and 2019. Further information about collection date, study name and treatment received are available in Supplementary Table 3. Samples were selected based on their *pfk13*, *pfmdr1*, *pfpm2* and *pfcrt* genotypes according to the following classification: sensitive, characterized by *pfk13* WT, *pfmdr1* monocopy, *pfpm2* monocopy, and no mutations at *pfcrt* codons known to modulate PPQ susceptibility (amino acid positions 88, 93, 97, 145, 343 and 353); KEL1/PLA1, a genetically defined group of *P. falciparum* parasites that emerged in the Greater Mekong Subregion, characterized by *pfk13* C580Y and *pfpm2* amplification[25], with or without polymorphisms at the aforementioned *pfcrt* codons; and MQ-R, characterized by *pfk13* mutation, *pfmdr1* amplification, and no polymorphisms at the aforementioned *pfcrt* codons. Characteristics and in vitro results of the selected samples are summarized in Supplementary Table 3. Genetic characterization was performed using the following methodology: polymorphisms in the *pfk13* propeller-encoding domain (PF3D7_1343700, codon 445 to 680) were determined by Sanger sequencing, as described by Ariey et al.[3]; *pfmdr1* and *pfpm2* copy number variations were determined by quantitative PCR according to Witkowski et al.[4], with the hybridization temperature adjusted to 63 °C. The presence of *pfcrt* (PF3D7_0709000) polymorphisms associated with PPQ resistance (codons 88, 93, 97, 145, 343 and 353) was determined by Sanger sequencing using primers described in Mairet-Khedim et al.[19] or whole genome sequencing data. Primers used can be found in the Supplementary Data 1. These isolates were adapted to in vitro continuous culture at 2% haematocrit (O+ human blood, Centre de Transfusion Sanguine, Phnom Penh, Cambodia) in RPMI 1640 supplemented with 0.5% AlbuMAX II, 2.5% human plasma (mixed serogroups), 5% $CO_2$ and 5% $O_2$, at 37 °C[19].

### Phenotypic assays

PPQ tetraphosphate (provided by the Worldwide Antimalarial Resistance Network (WWARN) and prepared in 0.5% lactic acid) was used for PSA, as previously described[4]. Briefly, *P. falciparum* parasites at the ring stage, 0–3 h post-invasion, were obtained by centrifugation in 75% Percoll and their concentration adjusted between 0.5% and 1% of parasitaemia. Parasites were incubated with 200 nM PPQ or 0.5% lactic acid (drug-free control). The drug was washed off after 48 h of exposure. Parasitaemia was measured in 10,000 red blood cells under a microscope, after 72 h of culture. For each *P. falciparum* strain tested, the survival rate was determined as the ratio between the parasitaemia in the drug-exposed condition and in the drug-free control condition. MQ (provided by the WWARN and dissolved in dimethyl sulfoxide (DMSO), Sigma Aldrich, Singapore) was used for in vitro MQ susceptibility in *P. falciparum* isolates using the [$^3$H]-hypoxanthine uptake inhibition assay, as previously described[19]. Briefly, parasites were synchronized at the ring stage by incubation with 5% D-sorbitol (0–12 h post-invasion) and exposed to MQ at different concentrations (2 to 1500 nM) in the presence of 0.5 μCi of [$^3$H]-hypoxanthine (PerkinElmer, Waltham, USA) for 48 h. Tritium incorporation was measured with a β-counter (MicroBetaTriLux; PerkinElmer Waltham, USA). Susceptibility to MQ was also measured using a survival test similar to the PSA, in which PPQ was replaced by MQ at defined concentrations, allowing the determination of a Survival $IC_{50}$ for MQ (defined as the drug concentration inhibiting 50% of parasite survival). $IC_{50}s$ and Survival $IC_{50}s$ were determined using the ICEstimator software (http://www.antimalarial-icestimator.net/) or the non-linear regression analysis tool in GraphPad Prism 7.0. Similarly, in vitro co-susceptibility to PPQ and MQ was assessed using a survival assay in 0–3-h ring-stage parasites, which were exposed to a combination of seven increasing concentrations of PPQ and MQ (ranging from 12.5 to 800 nM for each drug) for 48 h. Following drug removal, parasite cultures were maintained for an additional 24 h in drug-free medium. After a total of 72 h, Giemsa-stained blood smears were prepared, and *P. falciparum* survival was quantified as the ratio of viable parasites in the drug-exposed condition relative to the drug-free control. This allowed to determine a Survival $IC_{50}$ for the MQ–PPQ combination.

### Paired MQ–PPQ in vitro selection pressure

A *P. falciparum* strain (9097), collected in western Cambodia (Pursat) in 2019, was selected for continuous in vitro exposure to MQ and PPQ. This strain was isolated from a patient who experienced a late treatment failure on day 34 after receiving the recommended ACT (artesunate–mefloquine), as showed in Supplementary Fig. 5. Genomic characterization of the recrudescent parasite showed a duplication of the *pfmdr1* gene, associated with increased MQ $IC_{50}$ as determined by [$^3$H]-hypoxanthine incorporation.

Baseline genotypic analysis of strain 9097 identified the *pfk13* Y493H mutation associated with ART-R, single-copies of *pfmdr1* and *pfpm2*, and the absence of mutations in *pfcrt* on codons known to modulate PPQ susceptibility (amino acid positions 88, 93, 97, 145, 343 and 353). Phenotypic characterization was consistent with these genotypes, showing a ring-stage survival assay (RSA) slightly above 1% and no baseline in vitro resistance to MQ, PPQ, or amodiaquine.

The culture conditions and drug concentrations used for selection are summarized in Supplementary Fig. 1. Briefly, strain 9097 was first cultured in the presence of 40 nM MQ + 40 nM PPQ until no viable parasites were microscopically detectable (*Pressure 1*). At this stage, the cultures were returned to drug-free medium until parasitemia reached 2% (*Pressure 1 recovery*). A second drug exposure of 40 nM MQ + 40 nM PPQ had no observable effect, so the drug concentration was increased to 60 nM for each compound (*Pressure 2*). Once no viable parasites were microscopically detected, the cultures were again returned to drug-free medium until full recovery (*Pressure 2 recovery*). This was followed by a third round of drug pressure at 60 nM MQ + 60 nM PPQ (*Pressure 3*), followed by a recovery period (*Pressure 3 recovery*). Finally, a fourth and final round of pressure was applied with 80 nM MQ + 80 nM PPQ (*Pressure 4*).

Between each round of drug pressure, parasites were cryopreserved, genotyped (*pfmdr1* and *pfpm2* CNV), and phenotyped (MQ $IC_{50}$ determined by [$^3$H]-hypoxanthine incorporation, PSA, and survival rates under MQ and PPQ co-exposure: Survival $IC_{50}$).

### Whole-genome sequencing

Genomic DNA extracted from both the parental and *Pressure 4 P. falciparum* strains were used for library preparation (QIAamp DNA Blood Mini Kit (QIAGEN, Courtaboeuf, France)). Libraries were prepared using the Illumina DNA Prep protocol (ref. 20060059), starting from 250 ng of high-quality genomic DNA, and sequenced in paired-end

mode (2 × 150 bp) on a Nextseq 500 (Illumina). Raw reads were aligned to the *P. falciparum* 3D7 reference genome (PlasmoDB release 39) using the BWA-MEM algorithm (Burrows-Wheeler Aligner; default parameters). Alignment files were processed with SAMtools (version 1.4), and genome-wide coverage statistics were assessed using Quali-map (version 2.2.1). Duplicate reads were removed with Picard Mar-Duplicates (version 2.26.10).

SNPs and indels were identified using a custom analysis pipeline. A pileup file containing information on matches, mismatches, insertions, deletions, and mapping quality was generated using the SAMtools *mpileup* function (version 1.13). This file was used as input for custom Python scripts for genomic variant detection. Comparative analyses between the parental and *Pressure 4* strains were performed using custom R scripts (version 4.1), allowing identification of mutations acquired under drug pressure.

To assess CNVs, we used PlasmoCNVScan, a read-depth-based strategy specifically optimized for *Plasmodium* genomes. Analyses were performed as described by Beghain and colleagues[26], with the standard parameters: a 6-nucleotide motif and analysis limited to the exome.

### PPQ incorporation assay

Tritium-labeled PPQ (PPQ-[$^3$H(G)]) was purchased from American Radiolabeled Chemicals, Inc. (USA) and prepared in pure ethanol according to the manufacturer's instructions. Synchronized tropho-zoites (21–24 hours post-invasion) were obtained by centrifugation on a 75% Percoll gradient, followed by treatment with 5% D-sorbitol to remove remaining ring-stage parasites. Cultures were adjusted to a hematocrit of approximately 4%, with a minimum parasitemia of 2.5%.

Trophozoites were pre-incubated for 10 min with MQ (25–750 nM, depending on the experimental conditions). Then, PPQ-[$^3$H(G)] was added to a final concentration of 200 nM, and parasites were incubated for 5 hours at 37 °C under controlled atmospheric conditions (5% $CO_2$, 5% $O_2$).

PPQ-[$^3$H(G)] incorporation was quantified by calculating the ratio of intracellular to extracellular radioactivity. After incubation, infected erythrocytes were lysed using a solution containing 0.015% saponin and 0.1% Triton X-100. The intracellular fraction (PPQ*$_{in}$) and extra-cellular fraction (PPQ*$_{out}$) were measured using a β-scintillation counter (MicroBeta TriLux, Perkin-Elmer, Waltham, USA). The level of PPQ incorporation was expressed as the PPQ*$_{in}$/PPQ*$_{out}$ ratio.

### Statistical analyses

Statistical analyses were performed with the GraphPad Prism 7.0 soft-ware. A *p*-value < 0.05 was considered statistically significant. The normality of each dataset was tested using the Shapiro-Wilk test. All statistical tests were two-sided. Depending on normality check, either the Mann-Whitney *U* test, paired or unpaired *t*-tests or Wilcoxon matched-pairs signed rank test were used for comparisons between two groups. Kruskal–Wallis test followed by Dunn's multiple compar-ison test, ordinary one-way ANOVA followed by Tukey's multiple comparison test or by Dunnett's T3 multiple comparison test were used when comparing more than two groups. Correlation was assessed using a Spearman correlation test. Statistical significance is indicated in the figures by asterisks, using the following standard notation: * for $p \leq 0.05$ (significant), ** for $p \leq 0.01$ (very significant), *** for $p \leq 0.001$ (highly significant), and **** for $p \leq 0.0001$ (extremely significant).

Survival IC$_{50}$ values for the MQ–PPQ combination were deter-mined for each *P. falciparum* strain using non-linear regression analysis in GraphPad Prism 7.0 and the ICEstimator software (http://www.antimalarial-icestimator.net).

### Ethical clearance

All *P. falciparum* isolates were collected during Therapeutic Efficacy Studies (TES) conducted in compliance with the ethical guidelines of the Cambodian National Ethics Committee for Health Research (identifiers: NECHR #086 (Therapeutic Efficacy Study 2017), NECHR #087 (Therapeutic Efficacy Study 2017), NECHR #092 (Therapeutic Efficacy Study 2019), NECHR #099 (Therapeutic Efficacy Study 2016), NECHR #136 (Therapeutic Efficacy Study 2016) and NECHR #106 (Therapeutic Efficacy Study 2018)) and WHO Western Pacific Regional Office (WPRO) Ethical Review Committee. Written informed consent was obtained from all study participants or their legal guardians.

### Reporting summary

Further information on research design is available in the Nature Portfolio Reporting Summary linked to this article.

## Data availability

The source data underlying Figs. 1a–d, 2a–c, 3, 4a, b, 5a, b, and 6, Supplementary Figs. 1 and 2 are provided as a Source Data file. Due to the large file size, the data underlying Supplementary Fig. 3 are avail-able from the corresponding authors upon reasonable request. The next-generation sequencing data generated in this study have been deposited in the European Nucleotide Archive database under acces-sion code PRJEB85790. Source data are provided with this paper.

## Code availability

All scripts used for data processing and analysis in this study are available on GitHub at https://github.com/Rcoppee/MQ-PPQ_pressure. A permanent archived version of the repository is available via Zenodo at https://doi.org/10.5281/zenodo.17283874.

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

## Acknowledgements
We are grateful to the patients who accepted to participate in the therapeutic efficacy studies and to the staff of the Cambodian Health Centers for their contribution. This study was supported by the Global Fund RAI3E initiative grant through the WHO (B.W.).

## Author contributions
Conceptualization: C.R. and B.W. Methodology: C.R., A.C., J.P., R.C. and B.W. Validation: C.R., A.C., M.M.K., R.C. and B.W. Formal analysis: C.R., A.C., F.A., R.C. and B.W. Investigation: C.R., A.C., M.M.K., N.Kh., N.Kl., S.K., S.S., R.E., C.Ke. and C.Ku. Resources: L.A., R.L., P.R. and B.W. Original draft preparation: C.R. and B.W. Review and editing of the manuscript: C.R., A.C., M.M.K., J.P., P.R., F.A., R.C. and B.W. Visualization: C.R., A.C., R.C. and B.W. Project administration and supervision and funding acquisition: B.W.

## Competing interests
The funders had no role in the study design, data collection and interpretation, or in the decision to submit the work for publication. None of the authors declared a financial conflict of interest. P.R. is a staff member of the World Health Organization. The authors are solely responsible for the views expressed in this publication, which do not necessarily represent the decisions, policies, or views of the World Health Organization.

## Additional information

¹Malaria Unit, Pasteur Institute of Cambodia, Phnom Penh, Cambodia. ²Laboratoire de Parasitologie-Mycologie, UR ESCAPE, Université de Rouen Normandie, Rouen, France. ³GENOM'IC, Université Paris Cité, CNRS, INSERM, Institut Cochin, Paris, France. ⁴Infectious Disease Epidemiology and Analytics G5, Department of Global Health, Institut Pasteur, Université Paris Cité, INSERM U1347, Paris, France. ⁵National Center for Parasitology, Entomology, and Malaria Control, Ministry of Health, Phnom Penh, Cambodia. ⁶Mekong Malaria Elimination Programme, WHO, Phnom Penh, Cambodia. ⁷INSERM U1344, MERIT IRD,

Université Paris Cité, Paris, France. [8]Service de Parasitologie-Mycologie, Hôpital Cochin, Paris, France. [9]Centre National de Référence du Paludisme, Laboratoire de Parasitologie-Mycologie, Hôpital Bichat-Claude Bernard, Paris, France. [10]Centre National de Référence Cryptosporidioses, microsporidies et autres protozooses digestives, Centre Hospitalier Universitaire de Rouen, Rouen, France. [11]Present address: Genetic and Biology of Plasmodium Unit, Institut Pasteur de Madagascar, Antananarivo, Madagascar. ✉e-mail: bwitkowski@pasteur.mg

