## [Peer Review file · Nature Communications]

Tolerance of *Plasmodium falciparum* mefloquine-resistant clinical isolates to mefloquine-piperaquine with implications for triple artemisinin-based combination therapies.

Corresponding Author: Dr Benoit Witkowski

Version 0:

Reviewer comments:

Reviewer #1

(Remarks to the Author)

The authors address a highly relevant and timely topic, evaluating optimal malaria treatment strategies in the current context of emerging drug-resistant *Plasmodium falciparum* strains. With triple artemisinin-based combination therapies (TACTs) increasingly considered for deployment, it is critical to understand the resistance landscape of the regions where such therapies may be implemented, as this will influence their effectiveness.

In this context, the authors seek to assess the activity of the combination of mefloquine (MQ) and piperaquine (PPQ) against different resistance profiles, to understand whether triple ACTs remain efficacious in settings where MQ or PPQ resistance is present. This is an important question with significant implications for malaria control strategies, particularly in regions such as Southeast Asia, where drug resistance is evolving dynamically.

Through their phenotypic assays, the authors present an important and concerning finding: in the context of MQ resistance, the MQ + PPQ combination shows increased tolerance in *in vitro* susceptibility assays using field isolates.

However, some points could benefit from further clarification and refinement. In particular, concerns arise regarding how susceptibility comparisons are made using two different types of measurements — IC50 versus PSA. The authors present MQ IC50, PPQ survival assay, and MQ + PPQ IC50 (Figure A). It would strengthen the manuscript if the authors could also perform a survival assay for the MQ + PPQ combination, to provide a more consistent comparison across measurements. It would also be helpful to know whether the results of the survival assay would be expected to follow a similar trend for the MQ + PPQ combination. In the Materials and Methods section (line 275), it is stated that a survival assay was performed for MQ, but no corresponding results are shown.

Furthermore, the authors may wish to consider performing a checkerboard isobologram assay, which could provide a more detailed understanding of MQ and PPQ interactions under resistant and sensitive genetic backgrounds (DOI: 10.1016/j.ijpddr.2021.09.005).

To investigate the potential for stepwise adaptation to MQ + PPQ combination therapy, the authors selected a clinical isolate with a genetic background initially sensitive to both MQ and PPQ (i.e., no *pfmdr1* or *pfpm2* copy number variations). Interestingly, this isolate originated from a patient who experienced recrudescence after artesunate-MQ treatment, which resulted in *pfmdr1* amplification. Under MQ + PPQ *in vitro* selection, the strain acquired *pfmdr1* amplification by the "Pressure 4" timepoint. Figure S2 shows the different selection timepoints for which IC50, survival assay, and genotyping were performed. It would be informative to know the status of *pfmdr1* copy number at the "Pressure 2" timepoint — was amplification already present at that stage? If so, this might indicate selection from an initially mixed population, already favouring MQ resistance.

Additionally, the rationale for the specific MQ and PPQ concentrations used during *in vitro* selection could be more clearly explained. Since these drugs differ in potency, using the same concentration for both might bias selection toward resistance to the more potent drug. Similarly, the clinical isolate used is approximately two-fold more sensitive to MQ and 22-fold more sensitive to PPQ compared to resistant strains (Supplementary Table 1). This inherent sensitivity difference could have

favoured the early selection of an MQ-resistant genetic background.

Regarding genetic mediators of MQ + PPQ adaptation, the authors propose that *pfmdr1* amplification is the primary driver of tolerance. This is an interesting finding, although it appears to contrast with several prior studies, including some by the authors themselves, which have shown an inverse relationship between *pfmdr1* amplification and PPQ susceptibility. The resistance mechanism proposed (Figure 7) suggests a bidirectional role for PfMDR1; however, the current literature generally supports unidirectional transport for ABC transporters, including PfMDR1, which is known to transport substrates from the cytosol into the digestive vacuole using ATP hydrolysis. Further discussion or clarification of this proposed mechanism in light of existing knowledge would greatly strengthen this section of the manuscript.

The radiolabeled PPQ accumulation assay presented provides an interesting indirect association with PfMDR1 activity. However, as currently designed, the assay does not clearly distinguish whether the observed differences are due to cytosolic or digestive vacuole accumulation, which somewhat limits its ability to directly support the proposed role of PfMDR1 in modulating PPQ response. The authors may wish to consider complementing this approach with a competition assay using a known PfMDR1 substrate such as Fluo-4, which is well documented to be transported by PfMDR1. Additionally, it would be useful to acknowledge that while PfMDR1-mediated transport of MQ is well supported (DOI: 10.1038/sj.emboj.7601203; DOI: 10.1186/s12936-015-0791-3), there is currently no direct evidence for PfMDR1 transport of PPQ.

Although the authors rightly focus on *pfmdr1*, the CNV analysis of the "Pressure 4" strain also identified amplification of a region on chromosome 5. This amplification has previously been associated with PPQ resistance (DOI: 10.1128/aac.01793-10). The current study and the prior one thus provide independent evidence implicating this region in PPQ resistance. A previous attempt to correlate this CNV with PPQ susceptibility in Thai clinical isolates (doi:10.1128/AAC.06350-11) did not detect this amplification, likely due to lower selection pressure at the time of sample collection. The authors should consider further discussing the potential relevance of this chromosome 5 amplification in the context of their findings, and how it may contribute to PPQ resistance alongside *pfmdr1* amplification. It would also be valuable to explore whether such amplification patterns have become more prevalent in recent clinical isolates, given evolving drug pressure in the region.

Minor comments:

- Table S1: Standard deviation/error should be provided for IC50 and PSA values. I also suggest adding *pfpm3* CNV information, as it was analysed and is relevant given its emerging role in DHA-PPQ resistance in Africa (DOI: 10.1038/s41467-025-57726-5).
- PSA was determined through microscopy slide readings. As this technique can be subjective, the Methods section should specify how many independent experiments and independent slide readings were performed.
- Figure S3:
 - oPanel A: Add resistance threshold.
 - oPanel B: Consider adjusting the Y-axis to better visualise the differences.
- Additional details regarding the IC50 assay for MQ + PPQ co-exposure should be provided in the Materials and Methods section.
- In the Results section (lines 107–111), the reported MQ IC50 for the 9097 isolate appears inconsistent with the value shown in Table S1 — please clarify.
- Figure 6: Only 7 assays are shown for PPQ exposure alone. If the experimental design involved one assay per clinical isolate, there should be at least 12 assays.
- Discussion: The sentence "In vitro susceptibility testing confirmed that sensitive and KEL1/PLA1 parasites were highly susceptible to MQ, whereas WT and MQ-R parasites remained susceptible to PPQ. Conversely, KEL1/PLA1 and MQ-R strains displayed increased tolerance to PPQ and MQ." appears contradictory and would benefit from clarification. In summary, this is a valuable and timely study addressing a highly relevant question for the malaria research and control community. The findings presented are important, and the manuscript would be further strengthened by addressing the points raised above. I commend the authors for their work and hope that these comments will help improve the clarity and impact of the study.

End of reviewer comments.

Isabel Veiga

Reviewer #2

(Remarks to the Author)

The manuscript by Roesch et al entitled "Tolerance of *Plasmodium falciparum* mefloquine-resistant clinical isolates to mefloquine-piperazine: implications for triple artemisinin-based combination therapy strategies" describes a mechanism by which MQ resistance confers cross-tolerance to the MQ–PPQ combination, therefore questioning the foundation of this combination. The rationale of the work is well described; the analysis and results follow a logical flow; the results are convincing and well sustained; and the manuscript is well presented and clear. The results have important implications for malaria control.

I only have minor comments.

1. Authors use Tolerance, resistance, and over-sensibilization in an exchangeable way. Do they mean the same or there are different connotations for each term? Please explain if there are differences in their meaning or alternatively they mean the same, so that the reader follows clearly the concepts.

2. Abstract and discussion: authors mention that parasites rapidly acquire MQ-PPQ tolerance. It would be informative to give a temporal range for this rapid acquisition of tolerance.
3. Explain acronyms (for example, PSA in line 83) and key reagents/parasites (*P. falciparum* 9097 strain) at first use.
4. Explain briefly (either in results or methods section) key experiments instead of referring to supplementary material, to help readers follow the work presented ("four successive rounds of drug pressure, as summarized in the supplementary figure 2", line 103).
5. Explain key information used in the discussion instead of referring to other publications. For example: Line 222: authors propose that the fitness cost and the persistence of the amplification detected in their study (not evaluated) is probably similar to what Preechapornkul et al. observed. What did they observe?
6. Line 233: "the phenotypic discrepancies observed between MQ-R parasites and the other groups": What other groups do authors refer to?
7. Line 328 (Methods): "To assess copy number variations (CNVs), we used PlasmoCNVScan, a read-depth-based strategy specifically optimized for Plasmodium genomes". IS there any reference that described the key parameters used by authors to assess CNV? It is not trivial. If there is no reference, please provide the key information.
8. Figures: asterisks are misleading. Some figures have * and *** (missing **); others have *, **, ***, ****, but the legend does not explain what they mean. It may be better to provide directly the p value in the figure instead of using asterisks.
9. Statistical tests: authors mainly use non-parametric tests (except in few cases). Would it be better to perform all the test in non-parametric format? Alternatively, present the output of the normality test (Shapiro-Wilk) to avoid ambiguity. Special attention to Figure 6 where the main parameter (median survival) was tested as parametric and non-parametric.

Version 1:

Reviewer comments:

Reviewer #1

(Remarks to the Author)

The authors adequately addressed the key questions raised.

Reviewer #1 (Remarks to the Author):

The authors address a highly relevant and timely topic, evaluating optimal malaria treatment strategies in the current context of emerging drug-resistant Plasmodium falciparum strains. With triple artemisinin-based combination therapies (TACTs) increasingly considered for deployment, it is critical to understand the resistance landscape of the regions where such therapies may be implemented, as this will influence their effectiveness.

In this context, the authors seek to assess the activity of the combination of mefloquine (MQ) and piperaquine (PPQ) against different resistance profiles, to understand whether triple ACTs remain efficacious in settings where MQ or PPQ resistance is present. This is an important question with significant implications for malaria control strategies, particularly in regions such as Southeast Asia, where drug resistance is evolving dynamically.

Through their phenotypic assays, the authors present an important and concerning finding: in the context of MQ resistance, the MQ + PPQ combination shows increased tolerance in in vitro susceptibility assays using field isolates.

We thank the referee for the careful and constructive review. We fully agree that the potential reduction of efficacy due to resistance patterns already circulating in specific settings is a crucial point. Please note that we are also willing to highlight the important potential for drug resistance selection of the combination studied here. Notably, we managed to reproduce the selection of mefloquine resistance (MQ-R) from an isolate collected in Cambodia using MQ+PPQ pressure, despite the hypothesis that this particular combination would significantly reduce such risk.

However, some points could benefit from further clarification and refinement. In particular, concerns arise regarding how susceptibility comparisons are made using two different types of measurements — IC₅₀ versus PSA. The authors present MQ IC₅₀, PPQ survival assay, and MQ + PPQ IC₅₀ (Figure A). It would strengthen the manuscript if the authors could also perform a survival assay for the MQ + PPQ combination, to provide a more consistent comparison across measurements.

We agree with the reviewer that a survival assay combining MQ and PPQ is the most informative approach. This was indeed our rationale during the study design, which is why we used this method. However, we acknowledge that this point was not clearly explained in the manuscript, and we have now clarified it in the Methods section, part “phenotypic assays” (lines 318-334). To reinforce this clarification, for each data point obtained using the survival assay, rather than [³H]-hypoxanthine incorporation, we now systematically use the term “Survival IC₅₀”. This terminology has been defined in the Methods section (line 324). Moreover, there is now a statement in the manuscript (lines 98-100 and lines 115-117) providing more rationale about this methodological choice.

It would also be helpful to know whether the results of the survival assay would be expected to follow a similar trend for the MQ + PPQ combination. In the Materials and Methods section (line 275), it is stated that a survival assay was performed for MQ, but no corresponding results are shown.

The sentence refers to experiments performed on the parental and the selected lines after continuous exposure to MQ+PPQ. We acknowledge this was unclear and have revised the manuscript accordingly. Figure 2A shows results of survival assay with MQ performed on both «Parental» and «Pressure 4» strains. As previously noted, the term “Survival IC_{50s}” is now shown in all panels of Figure 2.

Furthermore, the authors may wish to consider performing a checkerboard isobologram assay, which could provide a more detailed understanding of MQ and PPQ interactions under resistant and sensitive genetic backgrounds (DOI: 10.1016/j.ijpddr.2021.09.005).

We agree with the reviewer that an isobologram analysis for MQ and PPQ would refine our understanding of the observed antagonism. Such data exist, but to our knowledge, they were generated using laboratory strains such as 3D7 and K1, and demonstrated strong antagonism between MQ and PPQ (Davis et al., 2006 AAC DOI: 10.1128/aac.00177-06). To some extent, our data echo these findings, although we did not use the isobologram approach. We observed a marked abrogation of PPQ activity in presence of MQ. Moreover, considering the relative concentrations of both drugs (see point below) in clinical contexts, fractional dose combinations used in isobologram might not reflect in vivo pharmacology. Finally, generating such isobolograms would require several months to be completed (strains thawing, adaptation, etc.).

*To investigate the potential for stepwise adaptation to MQ + PPQ combination therapy, the authors selected a clinical isolate with a genetic background initially sensitive to both MQ and PPQ (i.e., no *pfmdr1* or *pfpm2* copy number variations). Interestingly, this isolate originated from a patient who experienced recrudescence after artesunate-MQ treatment, which resulted in *pfmdr1* amplification. Under MQ + PPQ in vitro selection, the strain acquired *pfmdr1* amplification by the "Pressure 4" timepoint. Figure S2 shows the different selection timepoints for which IC₅₀, survival assay, and genotyping were performed. It would be informative to know the status of *pfmdr1* copy number at the "Pressure 2" timepoint — was amplification already present at that stage? If so, this might indicate selection from an initially mixed population, already favouring MQ resistance.*

We agree that the evolution of *pfmdr1* and *pfpm2* copy number under pressures is informative. We monitored it regularly. Tables below shows their evolution and have been added in the manuscript as “**Supplementary Table 1**”.

Pfmdr1	Pfmdr1 Parental	Pfmdr1 Pressure 1	Pfmdr1 Pressure 2	Pfmdr1 Pressure 3	Pfmdr1 Pressure 4
rep1	1,050	1,174	1,517	1,895	1,790
rep2	1,176	1,138	1,710	1,833	2,025
rep3	1,061	1,289	1,543	1,816	1,915
rep4	0,979	1,028			2,219
rep5	0,976				2,101
rep6	1,078				2,022
rep7					1,918
rep8					1,787
rep9					1,890
Mean	1,053	1,157	1,590	1,848	1,963
Std Dev	0,074	0,108	0,105	0,042	0,143

Dunnett's multiple comparisons test	Below threshold?	Summary	Adjusted P Value
Pfmdr1 Parental vs. Pfmdr1 Pressure 1	No	ns	0,4481
Pfmdr1 Parental vs. Pfmdr1 Pressure 2	Yes	****	<0,0001
Pfmdr1 Parental vs. Pfmdr1 Pressure 3	Yes	****	<0,0001
Pfmdr1 Parental vs. Pfmdr1 Pressure 4	Yes	****	<0,0001

Shapiro-Wilk test	Pfmdr1 Parental	Pfmdr1 Pressure 1	Pfmdr1 Pressure 2	Pfmdr1 Pressure 3	Pfmdr1 Pressure 4
P value	0,4316	0,9563	0,2377	0,3932	0,6487

Brown-Forsythe test - P value	0,4665
Are SDs significantly different (P < 0.05)?	No

Pfpm2	Pfpm2 Parental	Pfpm2 Pressure 1	Pfpm2 Pressure 2	Pfpm2 Pressure 3	Pfpm2 Pressure 4
rep1	1,260	1,253	0,859	0,846	0,912
rep2	1,338	0,958	1,138	0,768	1,021
rep3	1,199	1,058	1,120	0,996	1,060
rep4	0,761	1,118			0,949
rep5	0,853				0,973
rep6	0,869				0,930
rep7					0,870
rep8					0,842
rep9					0,818
Mean	1,047	1,097	1,039	0,870	0,931
Std Dev	0,247	0,123	0,156	0,116	0,080

Dunnett's T3 multiple comparisons test	Below threshold?	Summary	Adjusted P Value
Pfpm2 Parental vs. Pfpm2 Pressure 1	No	ns	0,9862
Pfpm2 Parental vs. Pfpm2 Pressure 2	No	ns	>0,9999
Pfpm2 Parental vs. Pfpm2 Pressure 3	No	ns	0,513
Pfpm2 Parental vs. Pfpm2 Pressure 4	No	ns	0,7183

Shapiro-Wilk test	Pfpm2 Parental	Pfpm2 Pressure 1	Pfpm2 Pressure 2	Pfpm2 Pressure 3	Pfpm2 Pressure 4
P value	0,2244	0,9672	0,1101	0,6556	0,9405

Brown-Forsythe test - P value	0,0071
Are SDs significantly different (P < 0.05)?	Yes

Additionally, the rationale for the specific MQ and PPQ concentrations used during in vitro selection could be more clearly explained. Since these drugs differ in potency, using the same concentration for both might bias selection toward resistance to the more potent drug.

We initially tested MQ-PPQ as an equimolar preparation, which we maintained for pressure experiments. The concern raised by the referee regarding potential bias is valid. However, in the parental strain, we started with a concentration (40 nM) above the PPQ Survival IC₅₀ but below the MQ Survival IC₅₀, PPQ being more potent. Despite this, we observed selection of MQ resistance, not PPQ resistance. Hence, the observed resistance is unlikely to be a methodological bias.

While translating in vitro data to the clinical settings has limitations, PK/PD studies evaluating MQ and PPQ indicate similar T_{max} for both drugs, with MQ reaching approximately 2.5-fold higher concentrations. The unbound fraction of MQ and PPQ are roughly estimated at 1-2% and 1-3%, respectively. Based on these values, equimolar concentrations may be observed in some patients (Tajerzadeh and Cutler, 1993 Biopharmaceutics & Drug Disposition DOI: 10.1002/bdd.2510140109).

Similarly, the clinical isolate used is approximately two-fold more sensitive to MQ and 22-fold more sensitive to PPQ compared to resistant strains (Supplementary Table 1). This inherent sensitivity difference could have favored the early selection of an MQ-resistant genetic background.

The first part of our manuscript aimed to evaluate the efficacy of MQ-PPQ against clinical isolates carrying prevalent resistance profiles from Asia. We concluded that MQ-R isolates exhibited higher resilience. For the in vitro pressure experiments, we chose strain 9097, which has acquired MQ-R in vivo, to maximize our chances of observing resistance selection under MQ-PPQ pressure. Thus, this strain was selected deliberately.

*Regarding genetic mediators of MQ + PPQ adaptation, the authors propose that *pfmdr1* amplification is the primary driver of tolerance. This is an interesting finding, although it appears to contrast with several prior studies, including some by the authors themselves, which have shown an inverse relationship between *pfmdr1* amplification and PPQ susceptibility.*

We would like to clarify that we do not reconsider our previous conclusions regarding the inverse correlation between *pfmdr1* amplification and PPQ susceptibility. We understand the referee's concern about strain "Pressure 4" (which carries *pfmdr1* amplification) potentially contradicting this. However, this contradiction would only hold if PPQ entered the parasites similarly in monotherapy and in combination with MQ. Our data suggest that PPQ entry is inhibited in the presence of MQ, preventing its antimalarial activity. Moreover, PPQ alone remained more effective than combined with MQ against this strain. We did not observe clear acquisition of PPQ-R: the slight increase in survival remained below the resistance threshold, and key molecular markers (e.g. *pfpm2/3* amplification or *pfcr1* mutation) were not detected.

The resistance mechanism proposed (Figure 7) suggests a bidirectional role for PfMDR1; however, the current literature generally supports unidirectional transport for ABC transporters, including PfMDR1, which is known to transport substrates from the cytosol into the digestive vacuole using ATP hydrolysis. Further discussion or clarification of this proposed mechanism in light of existing knowledge would greatly strengthen this section of the manuscript.

We agree with the referee that bidirectional transport involving PfMDR1 has not been demonstrated in *P. falciparum*, and we acknowledge that our hypothesis is too speculative. Our initial hypothesis was based on PPQ import inhibition and MQ export through PfMDR1 assuming that the digestive vacuole (DV) is MQ's primary target.

However, other studies provide more robust conclusions, whereby MQ accumulates in the DV for detoxification but exerts its main action in the cytosol, particularly the 80S rRNA sub-unit (Rohrbach et al., 2006, The EMBO journal DOI: 10.1038/sj.emboj.7601203 – Wong et al., 2017, Nature Microbiology DOI: 10.1038/nmicrobiol.2017.31). This hypothesis aligns with observations such as increased FLUO4 incorporation in *pfmdr1*-amplified parasites that could be inhibited in particular by MQ. As a result, we have significantly revised Figure 7 and the discussion to present an alternative hypothesis (PPQ and MQ competition for DV unidirectional entry through PfMDR1) instead of our previous statements.

The radiolabeled PPQ accumulation assay presented provides an interesting indirect association with PfMDR1 activity. However, as currently designed, the assay does not clearly distinguish whether the observed differences are due to cytosolic or digestive vacuole accumulation, which somewhat limits its ability to directly support the proposed role of PfMDR1 in modulating PPQ response. The authors may wish to consider complementing this approach with a competition assay using a known PfMDR1 substrate such as Fluo-4, which is well documented to be transported by PfMDR1. Additionally, it would be useful to acknowledge that while PfMDR1-mediated transport of MQ is well supported (DOI: 10.1038/sj.emboj.7601203; DOI: 10.1186/s12936-015-0791-3), there is currently no direct evidence for PfMDR1 transport of PPQ.

Reiling and Rohrbach showed that MQ inhibits FLUO4 import into the DV, an important aspect we had omitted in our previous discussion (now addressed, lines 271-273). However, we believe that repeating such experiment would not bring additional insights, especially since FLUO4 is not chemically comparable to PPQ. Similarly, although direct evidence is lacking, PPQ transport via PfMDR1 is a plausible hypothesis, considering the major impact of *pfmdr1* amplification on PPQ susceptibility (Mairet-Khedim et al., 2024 JAC - 10.1093/jac/dkac403).

Although the authors rightly focus on pfmdr1, the CNV analysis of the "Pressure 4" strain also identified amplification of a region on chromosome 5. This amplification has previously been associated with PPQ resistance (DOI: 10.1128/aac.01793-10). The current study and the prior one thus provide independent evidence implicating this region in PPQ resistance. A previous attempt to correlate this CNV with PPQ susceptibility in Thai clinical isolates (doi:10.1128/AAC.06350-11) did not detect this amplification, likely due to lower selection pressure at the time of sample collection. The authors should consider further discussing the potential relevance of this chromosome 5 amplification in the context of their findings, and how it may contribute to PPQ resistance alongside pfmdr1 amplification. It would also be valuable to explore whether such amplification patterns have become more prevalent in recent clinical isolates, given evolving drug pressure in the region.

We agree that our discussion lacked a detailed analysis of the chromosome 5 amplification. We have revised this section accordingly, comparing the amplified segments with those reported in previous studies. We also specifically compared Eastman's results with ours. (Lines 234-254).

Our conclusion highlights a notable divergence between the results reported by Eastman *et al.* and our findings. In their study, the authors found that this amplification led to high PPQ resistance, impaired PPQ accumulation and increased susceptibility to MQ. In contrast, in our study, we detected no major changes in PPQ susceptibility or accumulation, although MQ susceptibility changed dramatically. Genetically, our report revealed a larger amplification encompassing *pfmdr1*, whereas in Eastman *et al.*, the *pfmdr1* region was de-amplified. Importantly, the amplification described by Eastman *et al.* has not been associated with clinical PPQ-R in Cambodia (10.1016/S1473-3099(16)30415-7). Finally, Eastman *et al.* also reported the acquisition of a mutation on *pfcr1* which is known to be critical for the PPQ-R phenotype. While these genetic differences (*pfmdr1* and *pfcr1*) may contribute to the observed discrepancies, we consider it too speculative to draw further conclusions at this stage.

We also added a **Supplementary Figure 4** that schematizes the amplified portion in chromosome 5 on Pressure 4 lineage and recent PPQ-R isolates (1 from the isolates included in the manuscript and 3 collected in Cambodia in 2019) compared to results from Eastman *et al.* and Veiga *et al.* The conclusion was that the observation from Eastman *et al.* were not verified in recent PPQ-R strains (absence of amplification in the chromosome 5).

Minor comments:

•Table S1: Standard deviation/error should be provided for IC50 and PSA values. I also suggest adding pfp3 CNV information, as it was analysed and is relevant given its emerging role in DHA-PPQ resistance in Africa (DOI: 10.1038/s41467-025-57726-5).

As suggested by the referee, standard deviations have been added for MQ IC₅₀s measured via [³H]-hypoxanthine incorporation. Unfortunately, standard deviation for PSA were unavailable, as those experiments were performed in monoplicates.

Pfp3 CNV information is available on Figure 3. As for *pfp2*, no variation was detected.

•PSA was determined through microscopy slide readings. As this technique can be subjective, the Methods section should specify how many independent experiments and independent slide readings were performed.

Slides were read once by experienced microscopists. To reduce subjectivity, a minimum threshold of 10,000 RBCs was used for parasitemia estimation. Comparison between the “Parental” and the “Pressure 4” strains were based on multiple biological replicates.

•Figure S3:

oPanel A: Add resistance threshold.

There is no established consensus for MQ resistance threshold in the literature, with values ranging from 30 nM (Ringwald *et al.*, 1990) to 120 nM (Price *et al.*, 2004), depending on the assay type. These thresholds were often derived from *ex vivo* assays using fresh isolates, and cannot be directly compared to our results.

oPanel B: Consider adjusting the Y-axis to better visualise the differences.

As suggested, we added the resistance threshold in Panel A, and adjusted the Y-axis of Panel B to a logarithmic scale to better visualize the differences. Zero values are now shown as 0.001 for log scale purposes.

•Additional details regarding the IC₅₀ assay for MQ + PPQ co-exposure should be provided in the Materials and Methods section.

Additional details regarding these experiments were added in the Methods section (lines 318-334), which were initially unclear. We thank the reviewer for highlighting this.

•In the Results section (lines 107–111), the reported MQ IC₅₀ for the 9097 isolate appears inconsistent with the value shown in Table S1 — please clarify.

MQ IC₅₀ values in Figure 2 (lines 115-124) correspond to Survival IC₅₀s, not the [³H]-hypoxanthine-based IC₅₀s shown in the Supplementary Table 3. All IC₅₀ values presented in Figure 2 were derived from a modified version of PSA, where 0-3 h rings were exposed to increasing drug concentrations (0-800 nM), with washing after 48 h and assay completion at 72 h.

•Figure 6: Only 7 assays are shown for PPQ exposure alone. If the experimental design involved one assay per clinical isolate, there should be at least 12 assays.

We apologize for the missing five values in the graph; they were displayed as “0” and have now been corrected to 0.001 to enable log-scale visualization.

•Discussion: The sentence "In vitro susceptibility testing confirmed that sensitive and KEL1/PLA1 parasites were highly susceptible to MQ, whereas WT and MQ-R parasites remained susceptible to PPQ. Conversely, KEL1/PLA1 and MQ-R strains displayed increased tolerance to PPQ and MQ." appears contradictory and would benefit from clarification.

We thank the referee for noticing that. The word “respectively” was added on line 211 for clarity.

In summary, this is a valuable and timely study addressing a highly relevant question for the malaria research and control community. The findings presented are important, and the manuscript would be further strengthened by addressing the points raised above. I commend the authors for their work and hope that these comments will help improve the clarity and impact of the study.

End of reviewer comments.

Isabel Veiga

Reviewer #2 (Remarks to the Author):

The manuscript by Roesch et al entitled “Tolerance of Plasmodium falciparum mefloquine-resistant clinical isolates to mefloquine-piperaquine: implications for triple artemisinin-based combination therapy strategies” describes a mechanism by which MQ resistance confers cross-tolerance to the MQ–PPQ combination, therefore questioning the foundation of this combination. The rationale of the work is well described; the analysis and results follow a logical flow; the results are convincing and well sustained; and the manuscript is well presented and clear. The results have important implications for malaria control.

I only have minor comments.

1. Authors use Tolerance, resistance, and over-sensibilization in an exchangeable way. Do they mean the same or there are different connotations for each term? Please explain if there are differences in their meaning or alternatively, they mean the same, so that the reader follows clearly the concepts.

We acknowledge that the terms used in the manuscript (“tolerance”, “resistance”, “over-sensitization”) were not always clearly defined and may cause confusion. Although these terms were not intended to carry the same meaning, we agree that clarification is necessary. The term “tolerance” was used in contexts where we observed a reduced drug susceptibility in vitro, without being able to definitely link it to in vivo or clinical resistance. This term could be interchanged with “in vitro resistance” in some instances, though we aimed to avoid overinterpretation. By contrast, the term “resistance” was only used when a clear association with clinical resistance is established (either through validated molecular markers or defined phenotypic thresholds). We agree that the term “over-sensibilization” could be confusing, this term only used once was replaced by the term “higher susceptibility” (line 266).

2. Abstract and discussion: authors mention that parasites rapidly acquire MQ-PPQ tolerance. It would be informative to give a temporal range for this rapid acquisition of tolerance.

We added a temporal notion by adding the comment «for four months» (line 36) in the abstract. This information was also added on line 229.

3. Explain acronyms (for example, PSA in line 83) and key reagents/parasites (P. falciparum 9097 strain) at first use.

Here is a list of acronyms used and their definition:

- Triple artemisinin-based combination therapies (TACTs) (line 29)
- mefloquine–piperaquine (MQ–PPQ) (line 31)
- artemisinin-based combination therapies (ACTs) (line 48)

- artemisinin (ART) resistance (ART-R) (line 48)
- half-maximal inhibitory concentration (IC₅₀) (line 90)
- Survival half-maximal inhibitory concentration (Survival IC₅₀) (line 98)
- copy number variation (CNV) (line 134)
- single nucleotide polymorphisms (SNPs) (line 144)
- digestive vacuole (DV) (line 273)
- Therapeutic Efficacy Studies (TES) (line 296)
- ring-stage survival assay (RSA) (line 347)

The strain identification number (ie 9097) correspond to the ID assigned by the investigators to a given filed strain after its adaptation to continuous culture.

4. Explain briefly (either in results or methods section) key experiments instead of referring to supplementary material, to help readers follow the work presented (“four successive rounds of drug pressure, as summarized in the supplementary figure 2”, line 103).

Details regarding the different rounds of drug pressures applied in the selection experiments have been clarified in the Methods section (lines 349-358).

5. Explain key information used in the discussion instead of referring to other publications. For example: Line 222: authors propose that the fitness cost and the persistence of the amplification detected in their study (not evaluated) is probably similar to what Preechapornkul et al. observed. What did they observe?

As requested, additional key information related to the references cited in the manuscript were added (lines 260-262).

6. Line 233: “the phenotypic discrepancies observed between MQ-R parasites and the other groups”: What other groups do authors refer to?

The paragraph mentioning “other groups” has been fully rewritten, and the term has been removed.

7. Line 328 (Methods): “To assess copy number variations (CNVs), we used PlasmocNVScan, a read-depth-based strategy specifically optimized for Plasmodium genomes”. IS there any reference that described the key parameters used by authors to assess CNV? It is not trivial. If there is no reference, please provide the key information.

The following reference (PMID: 27066902) and a brief description of the parameters used were added in the Methods section (lines 380-381).

*8. Figures: asterisks are misleading. Some figures have * and *** (missing **); others have *, **, ***, ****, but the legend does not explain what they mean. It may be better to provide directly the p value in the figure instead of using asterisks.*

A description of the meaning of asterisks and their corresponding *p*-values was added in the “Statistical analyses” subsection of the Methods (lines 409-412). In addition, as suggested by the reviewer, exact *p*-values were added directly in the figure’s legends.

9. Statistical tests: authors mainly use non-parametric tests (except in few cases). Would it be better to perform all the test in non-parametric format? Alternatively, present the output of the normality test (Shapiro-Wilk) to avoid ambiguity. Special attention to Figure 6 where the main parameter (median survival) was tested as parametric and non-parametric.

All statistical results, including those from Shapiro-Wilk tests, have been included in the figure legends to help readers better understand the rationale behind the choice of statistical tests in each case.